# P=O Functionalized Black Phosphorus/1T-WS_2_ Nanocomposite High Efficiency Hybrid Photocatalyst for Air/Water Pollutant Degradation

**DOI:** 10.3390/ijms23020733

**Published:** 2022-01-10

**Authors:** Rak-Hyun Jeong, Ji-Won Lee, Dong-In Kim, Seong Park, Ju-Won Yang, Jin-Hyo Boo

**Affiliations:** 1Department of Chemistry, Sungkyunkwan University, Suwon 16419, Korea; jrh1015@naver.com (R.-H.J.); ljw9917@naver.com (J.-W.L.); weejusaen1@naver.com (S.P.); simpson94@naver.com (J.-W.Y.); 2Institue of Basic Science, Sungkyunkwan University, Suwon 16419, Korea; 3Thin Film Materials Research Center, Korea Research Institute of Chemical Technology, Daejeon 34114, Korea; Kimdongin200@naver.com

**Keywords:** 2D materials, hybrid visible photocatalyst, nanocomposite, plasma surface functionalization, environment pollutant removal

## Abstract

Research on layered two-dimensional (2D) materials is at the forefront of material science. Because 2D materialshave variousplate shapes, there is a great deal of research on the layer-by-layer-type junction structure. In this study, we designed a composite catalyst with a dimension lower than two dimensions and with catalysts that canbe combined so that the band structures can be designed to suit various applications and cover for each other’s disadvantages. Among transition metal dichalcogenides, 1T-WS_2_ can be a promising catalytic material because of its unique electrical properties. Black phosphorus with properly controlled surface oxidation can act as a redox functional group. We synthesized black phosphorus that was properly surface oxidized by oxygen plasma treatment and made a catalyst for water quality improvement through composite with 1T-WS_2_. This photocatalytic activity was highly efficient such that the reaction rate constant *k* was 10.31 × 10^−2^ min^−1^. In addition, a high-concentration methylene blue solution (20 ppm) was rapidly decomposed after more than 10 cycles and showed photo stability. Designing and fabricating bandgap energy-matching nanocomposite photocatalysts could provide a fundamental direction in solving the future’s clean energy problem.

## 1. Introduction

The pollution of the environment from chemicals used in industrial processes is a major concern, and the use of these chemicals continues to increase worldwide. Among these pollution types, water pollution is becoming increasingly alarming as the industry further develops. It is also a fundamental cause that can lead to soil pollution. Methylene blue (MB), a cationic thiazine dye widely used in fields such as the chemical and biological industries, can cause various health problems, including vomiting, nausea, extreme sweating, restless breathing, eye irritation, and mental disorders. Synthetic dyes are most widely used in the textile, leather, pharmaceutical, and food industries [1]. The use of dye-rich wastewater in many manufacturing industries can cause many health and environmental problems [2]. Synthetic dyes are toxic even at low concentrations and must be removed from industrial wastewater before discharge to aquatic systems. Over the past decades, many researchers have tried to find various systems and technologies to remove these dyes from industrial wastewater. Typically, adsorption [3,4,5], coagulation [6], ultraviolet decomposition [7], oxidation and reduction treatments are used to remove dyes [8,9,10]. However, many of these technologies are expensive and pose secondary risks, including toxic and hazardous chemicals. Conversely, the decomposition of organic dyes through a photocatalyst does not generate secondary chemicals. It is harmless to nature, making it convenient for industrial applications [11,12]. Photocatalytic decomposition of MB has long been attempted using various materials [13,14,15,16,17,18,19]. Additionally, in recent years, there has been a growing interest in volatile organic chemicals (VOCs) that are harmful to air quality and human health. Long-term exposure to VOCs can cause respiratory problems, such as lung cancer. In addition, VOCs can greatly affect the increase in fine dust by accelerating the formation of secondary organic aerosols that change into particles under oxidized conditions through complex heterogeneous reactions.

The discovery of graphene, a hot issue in the field of materials science, has clearly attracted great interest worldwide and has had a tremendous impact in many fields [20,21,22,23]. In the past few years, research into 2D materials has greatly advanced [24,25]. Research into layered 2D materials is at the forefront of materials science [26]. Among these materials, layered black phosphorus (BP) plays a vital role in several applications [27,28,29,30]. Phosphorene exhibits a layer-dependent bandgap ranging from 0.3 to 1.7 eV for monolayers in bulk [31]. Moreover, phosphorene is advantageous for optical applications because the bandgap depends only on the number of layers [32]. In addition, phosphorene has demonstrated high carrier mobility [33], layer-dependent photoluminescence (PL) [34], and anisotropic behavior [35]. Recently, Kim’s group showed that a bipolar pseudospin semiconductor could control the crystal structure of BP [36]. However, one major obstacle that phosphorene still suffers from is long-term stability in ambient conditions, and methods to overcome this through layer passivation or layer functionalization are being studied [37,38,39].

Contrary to these widely studied obstacles, oxidized BP remains largely unexplored [40]. Black phosphorus oxide can be produced using BP with differing oxygen content (P_x_O_y_). Theoretical calculations predict a wide bandgap range with increased oxygen concentrationfor BP oxide [41]. There have also been studies to improve battery performance by using P=O bonding as a redox functional group [42] and studies to dramatically improve the performance of reduction catalysts [43]. Tungsten disulfide (WS_2_), an important member of the layered TMD compounds, is widely applied in lubrication [44], field-effect transistors [45], electrocatalysts [46], and photocatalysts [47,48]. WS_2_ is a nonprecious metal–semiconductor material with a broad spectrum response and isused as a photocatalyst. In particular, single- and multiple-layer WS_2_ can be obtained through ultrasonic treatment or hydrothermal [49] or chemical vapor deposition [50] and exhibits excellent catalytic activity for visible light photocatalysts. It is often used as a cocatalystalong with other semiconductor photocatalysts [51,52]. Polymorphs of MX_2_ compounds, such as MoS_2_ and WS_2_, are usually ofseveral types, including 1T and 2H phases, based on the M and X atoms’ different coordination modes. The 2H phase can be described as consisting of two S-M-S layers composed of edge-shared MS6 triangular prisms. The 1T phase is characterized by one S-M-S layer composed of edge-shared MS6 octahedrons [53,54,55]. Compared withthe 2H phase with semiconductor characteristics, the 1T phase exhibits metallic characteristics and has high electrical conductivity, showing excellent performance [56]. However, the metal-phase WS_2_ nanostructure, especially the 1T phase, has been much less explored. Recent colloidal chemical synthesis has been proven impressive. It retains the stability of the synthesized metal-phase MoS_2_ over three months, which is much longer than 12 days of chemically exfoliated MoS_2_ that can be converted back from 1T to 2H [49]. Gopannagari et al. reported the composite of several layers of WS_2_ nanosheets and CdS nanorodscans [57]. The lifetime of the photogenerated carriers was effectively extended, and the surface migration properties were improved through the active edge sites and the intrinsic electrical conductivity around them. Therefore, the formation of heterojunctions is an effective way of improving photocatalytic performance [58]. These complex catalysts are designed to be bonded to each other, so that the band structure is designed for the purpose and can compensate for each other’s shortcomings. Among TMDCs, WS_2_ can be a promising cocatalyst material due to its unique electrical properties [59], and black phosphorus, which has been properly regulated for oxidation, can act as a redox functional group [42,43].

In this study, a photocatalyst for decomposition of water and air pollution was made through WS_2_ and composite materials by synthesizing black phosphorus, which is inexpensive, suitable, and easily controlled in oxidation state. A hybrid photocatalyst that satisfies the bandgap alignment and shortcomings through a nanocomposite material was studied. In addition, it was utilized as a functional group for P=O bonding, thereby having an additional effect of increasing efficiency and exhibiting improved catalytic efficiency to prevent electron-hole recombination. In addition, in terms of stability, which is a chronic problem of BP, the passivation effect was obtained with WS_2_ composite, and it was greatly improved. A synergistic effect was obtained from the excellent performance of the two materials, which are promising two-dimensional materials, and stability was also secured. In the field of two-dimensional materials, visible light photocatalytic materials may be additionally promising in various fields such as hydrogen generation and air and water purification. It can also be a way to overcome the limitations of BP with excellent performance.

## 2. Results and Discussion

### 2.1. Material Morphology Characterization

First, the oxidation state of BP was observed using SEM. The images in Figure 1 are samples before exfoliation, so several sheets are stacked. The exfoliated BP was measured using the sample before exfoliation to avoid electron beam damage and because it is difficult to observe with SEM. It observed that the initially cleaned bare BP is the same as the sheet-shaped commercial BP and that the surface is clean (Figure 1a,b). In addition, It observed that the surface is smooth even in high-magnification images. As the sample in this state is oxidated, P=O bonding occurs with increased amounts of P_x_O_y_ on the surface. When the sample is washed with water, P_x_O_y_ dissolves and disappears, leaving only partial P–O and P=O bonding. Figure 1c–f are the results of 5-W O_2_ plasma treatment for 5 and 10 min, respectively. This confirmed that the surface oxidation proceeds and is covered with P_x_O_y_, as can be seen clearly. At this time, in addition to the generation of the P_x_O_y_ layer, P–O bonding and P=O bonding are also formed [50].

Figure 2 shows the data obtained by measuring 1T-WS_2_ nanosheets synthesized with oxidation controlled BP (OBP) using high-resolution transmission electron microscopy (HR-TEM). HR-TEM measurements were obtained by dispersion in ethanol at 10 mg L^−1^ sample concentration and sampling in TEM 200-mesh Cu-carbon holey grid.

Figure 2a–c shows OBP TEM images. Wide and thin sheets can be seen at low magnification, and the appearance of the aligned grid structure at high magnification is clearly seen. This sample is partially oxidized. The BP can be damaged by a strong beam, so it was measured by lowering the laser power specifically. The observed high-resolution BP lattice distance is approximately 1.6 Å, which is confirmed in the BP’s *z*-axis direction [51,52]. It can be seen that the crystalline lattice is preserved even after the exfoliation of BP flakes, and there is a partial oxidation area. Figure 2d–f shows the HR-TEM images of the synthesized 1T-WS_2_ nanosheet. In the low-magnification image of Figure 2d, the shape of the thin nanosheet WS_2_ can be confirmed, and the lattice structures of 1T-WS_2_ are predominantly observed in the high-resolution image at high magnification. In Figure 2f, in the case of 1T-WS_2_, the six S atoms closest to the W atom are covalently bonded in a hexagonal shape [53]. In addition, the distance between the W–W atoms is as much as 3.1 Å in a uniform lattice structure.

Figure 3 shows the TEM and EDS data of the 1T-WS_2_ nanocomposite synthesized with the BP in the controlled oxidation state. As shown in Figure 3a, when viewed under low-magnification HR-TEM, two materials in the form of a sheet are combined. On average, the area of the WS_2_ sheet was often smaller than that of the BP sheet, and the ratio was formed randomly. Figure 3b,c shows HR-TEM images under high magnification, and both the BP and 1T-WS_2_ lattices seen in Figure 2 are observed. The observed samples had a random size through the exfoliation process, and WS_2_ sheets standing vertically in several layers were also observed. After checking the distribution of P, S, and W elements through TEM EDS mapping in Figure 3d–f, it was confirmed that the BP and WS_2_ nanosheets were bonded. Compared with the synthesized 1T-WS_2_, the BP material forms a much flatter and wider sheet, so most of the smaller 1T-WS_2_ sheets are distributed on the BP sheet.

### 2.2. Material Characterization

In the FT-IR spectrum, it can intuitively observe the BP oxidation state. In Figure 4a, the FT-IR peaks are assigned to P–O bonds at 1027, 1132, and 1639 cm^−1^ and assigned to the symmetric stretching mode of P=O bonds [54,55]. In pure BP, the observed peak was very weak. Also, it shows that even after the composite of WS_2_ and BP, these bonds were the same (Figure 4b). No peaks due to P–O and P=O were found in the composite with pure BP. This result shows that OBP was successfully composited on 1T-WS_2_.

Figure 4c shows the Raman spectra of the 1T-WS_2_, OBP, and OBP/1T-WS_2_ nanocomposites. For BP, the peaks located at 363, 440, and 467 cm^−1^ were assigned to the out-of-plane phonon mode A_g_^1^, planar phonon mode B_2g_, and planar phonon mode A_g_^2^, respectively [56]. For the 1T-WS_2_ nanosheet, two notable peaks are observed at 352 and 418 cm^−1^, which are assigned to the E_2g_ and A_1g_ modes, respectively [57]. Also characteristically, the peaks located at 131, 188, 261, and 325cm^−1^ in the low-frequency region correspond to the J_1_, J_2_, A_g_, and J_3_ peaks, respectively [57]. On the other hand, in the case of 2H-WS_2_, peaks such as J_1_, J_2_, and J_3_ do not appear, and the two main modes E_2g_ and A_1g_ also show a blue shift. The out-of-plane A_1g_ mode of 1T-WS_2_ is caused by the opposite oscillations of the two S atoms to the W atom. The E_2g_ mode is related to the in-plane oscillations of W and S atoms in opposite directions. For 1T-WS_2_, the J1 mode occurs at 131 cm^−1^, which is the out-of-plane motion of each stripe of the W atom inside the zigzag chain and the in-plane shear oscillation of one stripe of atoms relative to the other atoms in the chain. J_2_ is the movement of two zigzag chains relative to each other, and the J_3_ mode (325 cm^−1^) tends to break each zigzag chain in two stripes with some off-plane components [58]. This is one of the factors proving that the synthesized material is 1T-WS_2_ and not 2H-WS_2_. As a result, in the case of the OBP/1T-WS_2_ composite, the E_2g_ and A_1g_ peaks and all peaks show a red shift compared with the existing 1T-WS_2_, and the A_g_^1^, B_2g_, and A_g_^2^ peaks also show a red shift compared with a single OBP. It shows that electronic interaction is possible [59].

XRD patterns were used to identify the difference between BP and OBP of WS_2_ nanostructure and composite material with 1T-WS_2_ (Figure 5a–c). First, the XRD patterns of pure BP have peaks at 2θ = 17°, 34.3°, and 52.5° due to the (0 2 0), (0 4 0), and (0 6 0) planes (JCPDS 21-1272) [60]. After oxygen plasma treatment, the flake data showed that the intensity ratios of the three peaks became similar, and the overall peak intensity was greatly reduced. This is due to the distribution of amorphous P_x_O_y_ material on the surface. In addition, the peak tended to shift more depending on the degree of oxidation (Figure 5b). The data (blue) measured after removing P_x_O_y_ in the washing process show that the ratio between peaks is similar to that of pure BP, but the overall peak is shifted. The XRD pattern of the synthesized 1T-WS_2_ sample is shown in Figure 5c. The XRD peaks in the 15–19° and 56–59° regions of 1T and 2H can be used to identify 1T-WS_2_ or 2H-WS_2_ [61]. In addition, 2H-WS_2_ peaks around 15°, and 1T-WS_2_ appears around 18.3° [62]. The crystallinity of the composite material was observed with the peaks of both materials shifted by a 0.2° high angle. Shifting is due to the interaction between the two composite materials. However, it is an energy level that is difficult to observe clearly on the XRD.

The core-level peaks of W4f_7/2_ and W4f_5/2_ in the XPS spectrum are an intuitive and efficient way to differentiate between 1T and 2H phases [63]. As shown in Figure 5d, the double peaks located at 31.8 and 33.8 eV are due to the W4f_7/2_ and W4f_5/2_ core levels of 1T-WS_2_, respectively. On the other hand, at 32.7eV (W4f_7/2_) and 34.7eV (W4f_5/2_), the two strong peaks are characteristic of W for 2H-WS_2_, and it can be confirmed that the 1T phase is dominant [64]. In addition, peaks can be observed at 36.2 eV corresponding to W5p_3/2_ and W4f_7/2_ for both WS_2_ samples, which can appear in the amorphous WO_x_− on the surface [63]. In the case of the composite with OBP, the overall binding energy shifted to high energy, which is due to the change in electron density through the nanocomposite, proving that they are bound together. For OBP, peaks at 129.6, 130.5, and 134.8 eV were detected, corresponding to P 2p_3/2_, P 2p_1/2_, and phosphorus oxide (P_x_O_y_), respectively [65]. P=O bonding appears at 133.8 eV [66]. In this case for OBP/1T-WS_2_, these characteristic peaks have shifted to higher binding energies (129.9 and 130.8, respectively). The P_x_O_y_ and P=O bonding did not show any peak shift, but were still present. This is because WS_2_ bonding only affected the P–P bonding, not the P=O bonding side of the surface. Generally, binding energy transfer in composites means a strong interaction between two components. In the OBP/1T-WS_2_ hybrid, the transfer of binding energy to BP and WS_2_ is due to the change in the electron density, which means that the two materials are bonding and interacting [67].

The UV-Vis absorption spectrum (Appendix A) appeared similar to previously reported results. The characteristic peaks of the 1T phase for WS_2_ at ~450, 525, and 625 nm for the 2H-WS_2_ represent the 2H phase but were not found here [68]. Pure BP nanoflakes are known to exhibit very broad absorption from UV to NIR and are indicated by black lines [69]. In contrast, 1T-WS_2_ nanoflakes have a main absorption spectrum in the visible region. When 1T-WS_2_ is hybridized with OBP, the spectral shape looks similar to that of the sum of BP and WS_2_, but the spectral shape looks similar to that of 1T-WS_2_. This is because 1T-WS_2_ dominates light absorption in the visible area more than BP. The characteristics of the 2H-WS_2_ and 1T-WS_2_ phases are very different. The 2H-WS_2_ single layer is a semiconductor with a direct bandgap of approximately 2 eV. In contrast, it has been reported that 1T-WS_2_ has properties close to those of metals [70]. More precisely, the 1T-WS_2_ dispersion effectively absorbs light over the entire spectral range, with a monotonic decrease in absorbance as the wavelength increases. In addition, using the scattered reflectance of 1T-WS_2_ (Appendix A) in a simple bandgap calculation (Appendix A) through Kubelka–Munk’s theory [71,72,73], it confirmed that the narrow bandgap was close to zero.
(1)$$\frac{k}{s}=F\left(R_{\infty }\right)=\frac{{\left(1-R_{\infty }\right)}^{2}}{2R_{\infty }}$$

The bulk of the general 2H-WS_2_ has a 1.4 indirect bandgap, and it is known to have a direct bandgap of 2.0 eV as it becomes a single layer. However, the synthesized sample has a very narrow bandgap because it is a 1T metallic WS_2_ [74]. It is also known that BP has a bandgap of 0.2 eV in bulk state, and the bandgap increases with thickness. It was confirmed that the BP in the bulk state had a bandgap of 0.2 eV, and in the sample measured after exfoliation, it was measured to be about 1.7 eV.

### 2.3. Water Pollutant Photocatalytic Degradation

The MB decomposition photocatalytic activity of the previously investigated materials was evaluated using a solar simulator light source (wavelength 400–1000 nm). The change in the UV-Vis spectrum of the MB solution with photocatalytic reaction time is shown in Figure 6. Pure BP and OBP and synthesized 1T-WS_2_ and OBP/1T-WS_2_ composite materials were measured. The concentration of MB can be quantitatively determined as an absorbance peak at 665 nm using UV-Vis spectroscopy [75]. It conducted irradiation using only light (i.e., without a photocatalytic material) and observed for 1 h to confirm the photodecomposition of MB, and as a result, little appeared (Appendix A). In addition, it was stirred for 30 min in the dark to confirm the decrease in concentration due to the adsorption of the photocatalytic material. Usually, the initial adsorption of MB occurs relatively quickly, so the light source switch is turned on after checking the adsorption amount for 30 min. As a result, the amount of initial adsorption was negligible in pure BP, with an insignificant amount of less than 3%. OBP’s adsorption was also slightly but insignificantly increased. On the other hand, in the case of 1T-WS_2_, the amount of adsorption was relatively large because the specific surface area of the sample in vertical or wrinkled form even after exfoliation is high [76]. When looking at the concentration change after the start of the photocatalytic reaction, the photocatalytic efficiency of OBP increased slightly to more than that of pure BP, and 1T-WS_2_ showed satisfactory efficiency. However, there was an explosive increase in efficiency in the composite material. Also, the composite of pure BP and 1T-WS_2_ was investigated and showed better efficiency than when 1T-WS_2_ existed alone (Appendix A).

To summarize, there is a graph of MB decomposition over time in Figure 7a. The amount was adsorbed in the dark until 0 min. When the BP substance was present alone, the decomposition capacity of 20% for 60 min was limited even after oxidation treatment. The 1T-WS_2_ material showed an efficiency of 50% reduction in about 30 min. The composite material with pure BP showed an efficiency of 90% reduction in 30 min. The efficiency of the OBP/1T-WS_2_ material showed an almost 100% decomposition in 20 min. This is shown in Figure 7b for kinetic data for MB degradation, and these data were arranged to follow the pseudo-first-order response, as shown in Equation (2) [77,78,79].
(2)$$A_{t}=A_{0}e^{-\mathrm{kt}}$$

In Equation (2), k is the kinetic rate constant (min^−1^), A_0_ is the initial MB concentration, and A_t_ is the MB concentration after a certain time (t). That is, the kinetic rate constant can be obtained from the slope of the fitting line for a specific sample value. As a result, the BP with a very low photocatalytic effect was calculated as 0.51 × 10^−2^ min^−1^ and the OBP as 0.7 × 10^−2^ min^−1^. The kinetic rate constant of 1T-WS_2_ was 3.5 × 10^−2^ min^−1^. In comparison, the kinetic rate constant of BP/1T-WS_2_ nanocomposite was greatly improved to 6.94 × 10^−2^ min^−1^, and a very high value of 10.31 × 10^−2^ min^−1^ for OBP/1T-WS_2_ was calculated. These figures show much higher efficiencies than the reported types of MB decomposition photocatalysts [80,81,82] (Appendix A) and are measured without additional reducing agents such as NaBH_4_. The most important point is excellent stability. The capacity of the catalyst is very large, and even if the photocatalytic reaction is repeated 10 times, almost 100% of the organic dye is decomposed within that time.

Appendix A shows the TEM, XRD, FT-IR, and XPS data after 10-cycle photocatalytic reactions. As a result, it can see 1T-WS_2_ randomly placed on the BP sheet as before, and the lattice structure also exists. In the case of XRD, there was a slight decrease in intensity, but it was negligible. The same data were obtained after removing the MB adsorbed by van der Waals on the surface in FT-IR by several washings. Also, it was confirmed that there was no change in the material after the catalytic reaction, as XPS also showed no change in the peak location, except that the oxide area was slightly wider.

### 2.4. Air Pollutant Photocatalytic Degradation

The photocatalytic decomposition experiment of VOC, an air pollutant, was carried out with three substances. Experiments with BP, 1T-WS_2_ and OBP/1T-WS_2_ nanocomposites were each performed under sample conditions of 20 mg. In the case of blank, 200 ppm of acetaldehyde mixed gas was continuously injected and real-time check was performed by GC every 10 min. In the case of BP, both adsorption and decomposition amounts were insignificant. Meanwhile, 1T-WS_2_ adsorbed to some extent until the initial 100 min, but its decomposition into CO_2_ was poor. In Figure 8a, you can see that the amount of CO_2_ produced by decomposition is very insignificant. On the other hand, in the case of the OBP/1T-WS_2_ nanocomposite, after initial adsorption, VOC decomposition performance compared to the amount of sample was high, and even after 10 h, it showed steady efficiency. In theory, 1 mol of CH_3_CHO (acetaldehyde) can react with 5/2 mol of O_2_ to produce 2 mol of CO_2_.
2CH_3_CHO + 5O_2_ → 4CO_2_ + 4H_2_O

In Figure 8a,b, adding the amount of VOC output and CO_2_ conversion shows that it is similar to the initial amount of VOC input. Figure 8c shows the VOC consumption measured every 10 min. VOC consumption can be thought of as surface adsorption + CO_2_ conversion amount. Comparing the graph showing the amount of photocatalytic decomposition and converted into CO_2_ in Figure 8d, it can be seen that the initial consumption of VOC is large because the amount of adsorption is large. The decomposition rate of 1T-WS_2_ is almost insignificant, but the decomposition rate of OBP/1T-WS_2_ is 1.2 µmol/min. It is considering that the sample volume is 20 mg, the calculation is quite high with an efficiency of 60 µmol/g min. This calculated value can be said to be a very breakthrough value compared to the reported efficiency of other acetaldehyde photocatalysts (Appendix A) [83,84,85].

### 2.5. Efficiency Improvement Mechanism

In general, a typical semiconductor-based photocatalytic process proceeds as follows. Initially, light illumination induces an electron transition from valance band to conduction band, creating an electron–hole pair [86]. Then, the photo excited electrons and holes move to the surface and cause a reduction or oxidation reaction, respectively. However, electron–hole pairs recombine in the form of heat or emitted light. Therefore, separation and transport of electrons and holes are crucial factors in this reaction. The recombination of divided electrons and holes greatly influences the photocatalytic reaction and is directly related to the catalytic performance [87,88]. BP has excellent light absorption capability, but many studies have found that it is unfavorable for photocatalytic reactions because of the very rapid recombination [67]. Zhu et al. determined that it took several femto seconds for the separated electron–hole of pure BP to recombine using time-resolved diffuse reflection spectroscopy [89]. Thus, the BP band position is similar to that of 1T-WS_2_, and when a material with a narrower band is composited, the bands are aligned, and the donor receptor acts as a photocatalyst. Because the conduction band of 1T-WS_2_ is lower than that of BP, the electrons and holes excited in BP do not recombine and are transferred to 1T-WS_2_ to further form dye decomposition radicals. A schematic diagram of this process is shown in Figure 1. The stability problem of exfoliated BP is a major issue, and it expect that the oxidized P=O bonding and the binding of WS_2_ would help in terms of stability [90]. WS_2_ flakes can also provide a passivation layer effect. Many studies have already been reported on the effect of introducing passivating substances to solve the stability problem of BP [27,28,29,91]. Here, in order to clearly present the increase of the OH radicals generated on the surface, a fluorescence technique using the OH selective reaction of terephthalic acid, which has been reported before, was utilized [92,93]. An initial amount of 5 × 10^−4^ M terephthalic acid was dissolved in 2 × 10^−3^ M NaOH, and the fluorescence was measured by exiting at 315 nm by PL spectroscopy every 10 min. In this experiment, 10 mg of sample was added to 50 mL of terephthalic acid solution. In Appendix A, as a result of measuring the generation of OH radicals with each of the samples, there was no change in the peak at 425 nm in OBP, and there was a slight increase in the case of 1T-WS_2_. As shown in Appendix A, as a result of measuring the generation of OH radicals with each of the samples, as expected, there was no change in the peak at 425 nm in OBP, and there was a slight increase in the case of 1T-WS_2_. On the other hand, in the case of OBP/1T-WS_2_, it was confirmed that the peak caused by 2-hydroxyterephthalic acid was very high compared to the previous two samples. This is an indicator that the photocatalytic material can produce more OH in continuous time. It showed a tendency almost similar to that of the photocatalytic process using OH radicals, and it was dramatically improved in OBP/1T-WS_2_. In addition, H_2_O molecules adsorbed in the photocatalytic activity enhance the generation of OH radicals, but can suppress the photocatalytic decomposition activity by inhibiting the adsorption of decomposition substances [94]. However, in the case of OBP/1T-WS_2_, which already has O on its surface, rapid radical formation is possible without adsorption. This can be particularly beneficial for gaseous VOC decomposition processes.

EPR technology was used to further investigate the photogenic carriers and trapped electrons [95]. It is a technology that can be sensitively measured to investigate the unique electronic structure of photocatalytic materials and has been used to measure superoxide and hydroxyl radicals. The EPR spectrum is a device that observes the resonance absorption of microwave power by free radicals, and has obtained information about electron Zeeman interactions. Also, information about a sample with specific physical and chemical unfair electrons is a proportional constant called g-factor in the equation and varies with structure. Figure 9 shows the EPR spectrum for each sample. As expected, no distinct peaks were found in BP, which suggests that there are no unfair electrons or free radicals present [96]. In the case of 1T-WS_2_, it showed a strong and sharp EPR signal around g = 2.060, demonstrating the presence of radicals. In addition, in the case of OBP/1T-WS_2_, the g value slightly changed to 2.067, and the intensity of the peak was weakened, but sporadic peaks indicate the presence of various unfair electrons. Also, contrary to the symmetrical 1T-WS_2_ signal, it has an asymmetrical linear shape, which indicates that electrons in various states are irregularly present [97]. This is expected, as electrons trapped between the interface in the junction structure of two materials or electrons are generated by structural defects due to P=O [33]. These results suggest that it is an electronically advantageous environment for photocatalytic processes that are activated by radicals on the surface. Most of the previously studied data show a certain trend. Time-resolved photoluminescence spectra were measured to investigate the efficiency of the aforementioned charge separation. The TRPL measurements were performed in the linear domain, and the curve in the graph of Figure 9b shows that the TRPL attenuation of the composite material is much slower. Looking at the graph path, it was confirmed that the decay model for all samples was the same. As shown in Figure 9b and Table 1, it can be seen that the average electron lifetime of a single material is 5 ns, whereas the composite material is 7.3 ns, which is about a 50% improvement. Therefore, the photocatalytic reaction is more likely due to this longer electron lifetime. This can be said to correspond to the experimental results and mechanism shown above. Meanwhile, we utilized the P=O site because it offers molecular level control of the oxidized P active site. It is stabilized by covalent bonds at the interface and by contact with 1T-WS_2_, which has high electronic conductivity and metallic properties [98]. In addition, BP, which already has the presence of O_2_, a material for radical formation on the surface, has favorable conditions for radical formation efficiency [99]. P=O bonding, which can exist at the interface between BP and 1T-WS_2_, also affects the redox level, enabling smooth band alignment, resulting in high efficiency. In addition, the presence of P=O bonding increases the probability of trapping electrons at the interface, and since this functional group has a negative charge, it becomes easy to attract cationic dyes having a positive charge into the decomposition reaction. The basic photocatalytic mechanism applies equally to VOC reduction. In particular, catalyst efficiency in a gaseous state is the primary factor that affects the efficiency of effective adsorption. The vertical and corrugated structure of 1T-WS_2_ has a large specific surface area, so basic adsorption is very easy, and it has advantageous conditions [100,101]. Continuous accumulation of adsorbed molecules can also lead to deactivation of the adsorbents if it cannot be decomposed in a timely manner through the photocatalytic process [102]. However, according to previous VOC reduction experiments, the adsorbed VOC can be further decomposed into CO_2_, H_2_O or other intermediates after reaching adsorption equilibrium, which keeps the removal efficiency of the VOC remaining in the photocatalytic process at a stable high level. In addition to favorable adsorption conditions, the presence of P=O functional groups on the surface serves as a very favorable condition for VOC conversion requiring active O species. Most of the common VOC reduction photocatalysts were TiO-based materials, but it is meaningful that 2D-based VOC reduction materials were developed in this study.

## 3. Materials and Methods

### 3.1. Materials

Bulk BPcrystals were purchased from ACS Material (Pasadena, CA, USA). Tungsten chloride (WCl_6_), thioacetamide (C_2_H_5_NS), *N*-methyl-2-pyrrolidone (NMP),dimethylformamide (DMF), MB, and all other chemicals were acquired from Sigma-Aldrich without further purification.

### 3.2. Controlling the Oxidation of BP

BP powder was treated with 5W of oxygen plasma for 5–10 min with an oxygen flow of 100 sccm under 1.5 Torr in a plasma system (COVANCE, Femto Science Inc., Hwaseong, Korea).If BP’s oxidation treatmentis strong, the surface is etched, so the low wattage was fixed and adjusted only by time.

### 3.3. Preparation of 1T-WS_2_Nanosheet

1T-WS_2_ nanosheets were synthesized usingthe hydrothermal method. WCl_6_ (0.1 M) and thioacetmide (0.2 M) were added to 38 mL ofpurified water and stirred under vacuum for 1 h. The solution was put in a 50 mL Teflon autoclave and reacted at 210 °C for 8 h. The nanosheetswere then naturally cooled, washed with ethanol threetimes, centrifuged, and dried.

### 3.4. Synthesis of Oxidation Controlled BP/WS_2_Nanocomposites

The 20-mg 1:1 ratio BP/WS_2_ powder obtained after exfoliationwas added to a 50-mL NMP solution and sonicated for 36 h. Finally, the sample was obtained through high-speed centrifugation, washed thoroughly using ethanol, and redispersed in 10 mL of ethanol to obtain an oxidation-controlled BP (OBP)/1T-WS_2_ nanocomposite.The overall experimental process is shown in Appendix A.

### 3.5. Photocatalytic MB Degradation

The MB solution concentration in the experiments performed was 20 mg L^−1^. The catalyst (1 mg) was added to 20 mL of the MB solution. First, the MB solution was subjected to light for an hour to confirm the light decomposition and absorbance of the solution. After the addition of a photocatalyst, the mixture was stirred in the dark for 30 min. The adsorption amount was measured through absorbance. Later, light irradiation was started, and samples were taken every 5 min and centrifuged. The separated solution was transferred to a cuvette to measure absorbance. The light source used was a solar simulator (Model-DXP300, DY-tech Co., Copthorne, UK) (output wavelength: 350–1000 nm) with a UV cut-off filter. All supernatantsweremeasured using anultraviolet-visible (UV-Vis) spectrophotometer. The recycling test of the catalyst was repeated after the MB was completely decomposed. The process for this is shown in Appendix A.

### 3.6. Photocatalytic VOC (Acetaldehyde) Degradation

VOC photocatalytic activity tests (Appendix A) were measured using a quartz tube furnace equipped with gas chromatography (GC, GC, Hewlett Packard, Palo Alto, CA, USA, HP 6890). A temperature control in quartz tube furnace and was conducted at room temperature has been precisely control the flow rate of the gas flowing into Massflow controller (MFC, Toronto, ON, Canada). During the experiment, humidity and impurities in the quartz tube were controlled respectively. An additional gas path through which the dry air passes through the water bottle is installed to control the humidity of the gas mixture entering the reactor, and 20 mg of each catalyst uniformly dispersed in quartz wool prior to the acetaldehyde (CH_3_CHO) decomposition experiment is added to dry air (30 mL/min) under 20W white LED light source was monitored to remove impurities for 24 h under a constant flow. After that, a mixed N_2_gas containing 200 ppm of Acetaldehyde was injected into the reactor. The amount of CO_2_ and VOC in the gas mixture passing through the reactor was measured every 10 min via a GC connected to the reactor outlet. The catalytic activity was evaluated in terms of VOC consumption (ppm) and CO_2_ converted emissions (ppm). In theory, 1 mol of CH_3_CHO (acetaldehyde) can react with 5/2 mol of O_2_ to produce 2 mol of CO_2_.

### 3.7. Characterization

BPand WS_2_ are characterized usingfield-emission scanning electron microscopy (Model JSM-7100 F), which evaluates their surface morphologies. OBP, 1T-WS_2_, and OBP/1T-WS_2_ nanocomposite lattice structures and exfoliation status were observed usingCs-corrected transmission electron microscopy (Cs-corrected TEM, Model JEOL-JEM ARM 200F, Tokyo, Japan) at an accelerating voltage of 80–200 kV. This unit is equipped with EM, HAADF, FLC, UltraScancharge-coupled detector (CCD) camera, and EDSunits.The material crystallinity was characterized byX-ray diffraction (XRD, D/Max Ultima III, Rigaku Corporation, Akishima, Japan). Fourier transform infrared (FT-IR) spectroscopy was performed with a Bruker Optics, VERTEX 70 (Billerica, MA, USA). The spectra were recorded with a diamond-attenuated total reflection unit in a spectral range from 4000 to 300 cm^−1^. The surface composition and chemical state of the samples was evaluated by X-ray photo-electron spectroscopy (XPS) (Vg Scienta, Tonbridge, UK, ESCA 2000). The absorption spectrum of the material was measured using a UV-Vis absorption spectrophotometer (UV-3600 Plus UV-Vis-NIR spectrophotometer, Shimadzu Corporation, Kyoto, Japan). The MB’s catalytic decolorization was monitored using UV-Vis absorption spectroscopy (UV-3600 Plus UV-VIS-NIR Spectrophotometer, Shimadzu Corporation, Kyoto, Japan). Additionally, the MB concentrations and the catalyst capacity were both measured using UV-Vis absorption techniques. In the confocal Raman spectroscope (NTEGRA Spectra, NT-MDT Co., Zelenograd, Russia) measurements, a 532-nm wavelength laser was linearly polarized. The objective lens applied in all experiments has a 0.7 NA and 100× magnification (Mitutoyo, Japan). The Raman scattering signals were obtained through a CCD (Andor, UK) cooled to −75 °C and a spectrometer with a grating of 1800 grooves/mm, blazed at 500 nm. Electron Paramagnetic Resonance spectrometry (JEOL-JES-FA200 EPR spectrometry, Tokyo, Japan) was used to study material organic radicals. The measurement temperature was −150 °C and the measurement method was the CW-EPR method.

## 4. Conclusions

In this study, O_2_ plasma used to control the oxidation of BP and synthesized BP with P=O bonding. In addition, WS_2_ with dominant 1T phase was synthesized using the hydrothermal method. Two materials nanocomposite were synthesized was performed by the sonication method. This nanocomposite catalyst showed excellent photocatalytic performance against high-concentration MB decomposition in a solar simulator equipped with a UV cut-off filter (wavelength 400–1000 nm). The reaction rate constant *k* increased to 10.31 × 10^−2^ min^−1^, far superior to those of other reported photocatalysts. In addition, even though the photocatalytic reaction was carried out for up to 10 cycles, no decrease in efficiency or deterioration of material was observed. The improved photoactivity and stability were due to the wide absorption and recombination prevention, the fast conductivity of the two materials, and the effect of the redox functional group of P=O bonding. The stability problem of peeled BP was also solved by P=O bonding and bonding with 1T-WS_2_.In addition, it showed 4 ppm/min decomposition performance in reducing acetaldehyde, which is a gaseous pollutant VOCs. This was not photocatalytic activity in the BP and 1T-WS_2_ single material samples, but showed activity in the composite material. This study presented a method to solve the BP’s stability problem and the fast and dominant electron–hole recombination problem at the same time. We observed great potential and promisein applying 1T-WS_2_ in photocatalysis. In addition, the oxidation control of BP through plasma treatment can be expected to be applied to other materials. The problem of VOC-based fine dust is serious, so the need for an eco-friendly solution has increased greatly. The development of hybrid photocatalyst materials capable of decomposing both air and water pollutants can be a fundamental solution to the environmental pollution problem.

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
