# Peer review of "P=O Functionalized Black Phosphorus/1T-WS2 Nanocomposite High Efficiency Hybrid Photocatalyst for Air/Water Pollutant Degradation"

_ijms, 2022, doi:10.3390/ijms23020733_

Round 1
Reviewer 1 Report
In this work, the authors designed a composite catalyst that showed excellent photocatalytic performance against MB decomposition. The composite is formed of black phosphorus that was properly surface oxidized by oxygen plasma treatment and combined with 1T-WS2. The manuscript must be improved and carefully revised.
- First of all, the title is not well-written. It must be changed and corrected (pollutant instead of polluant, for example).
- The starting materials are not well characterized. AFM measurements are required to confirm the thickness of these materials.
- Some scale bars are too small and unreadable. In addition to this, the numbers of many graphs are too small.
- W element mapping is missing. Moreover, the 2H-WS2 Raman spectrum should be included.
- The authors should compare the photocatalytic performance of the nanocomposite with a physical mixture in order to clearly see the synergy of this hybrid material.
- Have the authors characterized the nanocomposite after the catalytic tests (SEM, TEM, XPS, etc)?
- XPS and FT-IR experiments are not commented in the experimental section.
- The introduction and conclusions should be improved. What is the novelty of this work?
- The included references in Table S1 and S2 are not correct. Please, check.
Author Response
To
The Editor -in -Chief,
International Journal of Molecular Science (IJMS)
Dear Sir,
Here with I am submitting the revised version of the article entitled “P=O functional black phosphorus / 1T-WS2 2D nanocomposite enhanced efficiency hybrid visible-light photocatalyst for environmental polluant degradation” by Rak Hyun Jeong and Jin-Hyo Boo modified according to the suggestions of the reviewers.
Modifications made in the present manuscript are listed below.
- Comments of the Reviewers have been answered (Next page: Replies to Reviewer’s comments) and incorporated in the manuscript.
- Title is changed.
- The manuscript content has been added. (1. Introduction, , 3.7 Characterization, 4.Conclusion)
- Correction of typo and English errors.
- Supplementary Data is added. (Figure S3, Figure S4 b-d)
- Correction of reference number. (Table S1,S2)
- All figures are resized.
With the above modifications and other corrections, the manuscript has been improved to a great extent. We believe that the present version will fulfill the expectations of the referees and meet the quality standards of the International Journal of Molecular Science (IJMS). Thank you so much for your interest and valuable suggestions to improve the manuscript. Please do not hesitate to ask questions if you do not have enough answers or if you have any additional questions. We will do our best to respond that as a top priority.
Kindly consider this manuscript for the publication in your esteemed journal.
Yours sincerely,
Jin-Hyo Boo.
Reply to the Reviewer’s comments
Reviewer 1
In this work, the authors designed a composite catalyst that showed excellent photocatalytic performance against MB decomposition. The composite is formed of black phosphorus that was properly surface oxidized by oxygen plasma treatment and combined with 1T-WS2. The manuscript must be improved and carefully revised.
- First of all, the title is not well-written. It must be changed and corrected (pollutant instead of polluant, for example).
Reply: I sincerely appreciate the comment of the reviewer. In response to the reviewer's opinion, the article title has been changed to be more concise and intuitive.
‘P=O functional black phosphorus / 1T-WS2 2D nanocomposite enhanced efficiency hybrid visible-light photocatalyst for environmental polluant degradation’
→ ‘P=O functionalized black phosphorus / 1T-WS2 nanocomposite high efficiency hybrid photocatalyst for air/water pollutant degradation’
- The starting materials are not well characterized. AFM measurements are required to confirm the thickness of these materials.
Reply: Thanks to kind instruction. Compared with the SEM photo before exfoliation, when looking at the state in the TEM data after exfoliation, the thickness was not measured separately because it was the state of the few layers. The thickness of all samples in this experiment is seems to be less than 20 nm. The thickness of the starting material was not considered to be an important factor, so it was excluded from the manuscript data. In this study, in the case of 1T-WS2, it was observed that it was in the form of a curved thin sheet rather than a flat sheet, so it was difficult to accurately measure the thickness by AFM measurement. The sensitivity of the AFM equipment was required because the samples were several nm thick. We apologize for not being able to measure within a short period of time with our university's equipment. We will make sure to check whether the estimated thickness of the starting material is correct.
- Some scale bars are too small and unreadable. In addition to this, the numbers of many graphs are too small.
Reply: We apologize for this and amended it immediately. The size of the figure is small, so it seems unreadable. Each figure has been resized to make it more visible.
- W element mapping is missing. Moreover, the 2H-WS2 Raman spectrum should be included.
Reply: Thanks to kind instruction. W element mapping was initially measured, but W was measured with L series, whereas P element mapping was measured with K series. Therefore, since the S element exists only in the form of WS2 in this sample, it was measured with the S element measured through the same K series. The W element mapping data did not differ from the S element. Other EDS mapping data is attached as reference. I've included a supplementary picture in the word file (reply to reviewers)
Also characteristically for 1T-WS2, the peaks located at 131, 188, 261, and 325cm−1 in the low-frequency region correspond to the J1, J2, Ag, and J3 peaks, respectively. On the other hand, in the case of 2H-WS2, peaks such as J1, J2, and J3 do not appear, and the two main modes E2g and A1g also show a blue shift. 2H-WS2 Raman spectra data is attached as reference.
I've included a supplementary picture in the word file (reply to reviewers)
- The authors should compare the photocatalytic performance of the nanocomposite with a physical mixture in order to clearly see the synergy of this hybrid material.
Reply: Thanks to kind instruction. When a photocatalytic experiment was conducted with two materials as a simple physical mixture, it was confirmed that the materials were dispersed in water by stirring and the efficiency was significantly reduced. It is difficult to expect a synergistic effect between the two substances because the contact itself is difficult. In fact, in the case of physical mixtures, only the result of adding up the photocatalytic performance by sample amount was obtained, so it was excluded from the result data. Also, the photocatalytic stability of the materials was not good.
- Have the authors characterized the nanocomposite after the catalytic tests (SEM, TEM, XPS, etc)?
Reply: I am so sorry to confuse the Referees. We apologize for this and amended it immediately. There was a problem in the process of modifying the supporting information. There was a mention in the manuscript, but data was missing in the supporting information. According to the contents of the manuscript, SEM, TEM, FT-IR, and XPS data after photocatalytic testing have been added to Figure S3.
- XPS and FT-IR experiments are not commented in the experimental section.
Reply: We apologize for this and added it immediately. According to the reviewer's opinion, the relevant content has been added. It's about highlighting manuscript.
- The introduction and conclusions should be improved. What is the novelty of this work?
Reply: Thanks to kind instruction. According to the reviewer's opinion, the relevant content has been added. It's about highlighting manuscript.
- The included references in Table S1 and S2 are not correct. Please, check.
Reply: I am so sorry to confuse the Referees. We apologize for this and amended it immediately. There was a problem in the process of modifying the supporting information. It's about highlighting manuscript.

Reviewer 2 Report
Authors present an interesting and complete study of P=O functional black phosphorus / 1T-WS2 2D nanocomposite as photocatalytic material. Here I have some comments for its consideration.
- The manuscript is long and sometimes is a bit hard to follow it. Please, consider a better structure for the manuscript, maybe dividing the Results and Discussion into multiple sections.
- Moderate English changes required (Co-catalyst/cocatalyst, bandgap/band gap, abuse of "we", etc.).
- Please, also revise chemical compounds naming (CO2, etc.)
- Why did the authors consider 20 ppm as a high concentration? There are available papers using 50 ppm of MB.
- Have you performed experiments in a continuous regime?
Author Response
Reviewer 2
Authors present an interesting and complete study of P=O functional black phosphorus / 1T-WS2 2D nanocomposite as photocatalytic material. Here I have some comments for its consideration.
Reply: First of all, we appreciate the positive feedback of the referee. We modified the part in the manuscript for your comments. Please do not hesitate to tell us if there is anything that you think is not being answered. Thanks to kind instruction.
- The manuscript is long and sometimes is a bit hard to follow it. Please, consider a better structure for the manuscript, maybe dividing the Results and Discussion into multiple sections.
Reply: Result and Discussion was further subdivided into multiple sessions. With this, we believe the article quality has been improved. Thanks to kind instruction.
- Moderate English changes required (Co-catalyst/cocatalyst, bandgap/band gap, abuse of "we", etc.).
We apologize for this and amended it immediately. We unified word notation as cocatalyst, band gap, and corrected the abuse of ‘we’. It's about highlighting manuscript. Thanks to kind instruction.
- Please, also revise chemical compounds naming (CO2, etc.)
We apologize for this and amended it immediately. Thanks to kind instruction. We have corrected the chemical compound naming such as ‘CO2’ mentioned by the reviewer. We have corrected all the points you pointed out and other typo. It was revised according to the reviewer's comments and highlighted that part.
- Why did the authors consider 20 ppm as a high concentration? There are available papers using 50 ppm of MB.
Reply: I am so sorry to confuse the Referees. In this experiment, the absorbance of 20 ppm MB solution is about 2.0. In general MB photocatalyst decomposition papers, absorbance is measured at about 1.0 (about 10 ppm), so 20ppm is indicated as high concentration. Adsorption and reduction catalysts are also measured at higher concentrations. There is a measurement limit of the equipment (UV-Vis spectrophotometer), so it was measured at a high concentration within the possible range. For this reason, we have referred to it as a relatively 'high concentration' in this paper.
- Have you performed experiments in a continuous regime?
Reply: Yes. Photocatalyst and recycle experiments were all carried out in a continuous regime.
